# Association between Initial Severity of Facial Weakness and Outcomes of Bell’s Palsy

**DOI:** 10.3390/jcm10173914

**Published:** 2021-08-30

**Authors:** Myung Chul Yoo, Dong Choon Park, Seung Geun Yeo

**Affiliations:** 1Department of Physical Medicine & Rehabilitation, School of Medicine, Kyung Hee University, Seoul 02447, Korea; famousir@naver.com; 2St. Vincent’s Hospital, The Catholic University of Korea, Suwon 16247, Korea; park.dongchoon@gmail.com; 3Department of Otorhinolaryngology, Head and Neck Surgery, School of Medicine, Kyung Hee University, Seoul 02447, Korea

**Keywords:** Bell’s palsy, initial severity, prognosis, favorable outcome

## Abstract

To establish whether clinical prognostic factor outcomes differed based on the initial severity of facial weakness and to determine the association between the initial severity of facial weakness and favorable outcomes. This retrospective cohort study analyzed all patients with Bell’s palsy who visited the outpatient clinic of our university hospital from 1 January 2005 through 31 January 2021. The primary outcome was the rate of recovery at 6 months, evaluated separately in patients with initial House–Brackmann (H-B) grades 3–4 and 5–6. Secondary outcomes included clinical factors associated with favorable outcomes stratified by the initial H-B grade. The rate of favorable recovery was higher in patients with initial H-B grades 3–4 than initial H-B grades 5–6 (82.9% vs. 68.2%, *p* < 0.001). Multivariable logistic regression analysis showed that age 19–65 years and good electromyography (EMG) results were prognostic of good outcomes in patients with initial H-B grades 3–4. In addition, good EMG results, controlled hypertension, and combination antiviral therapy were significantly prognostic of favorable outcomes in patients with initial H-B grades 5–6. Subgroup analysis interactions showed that combination antiviral therapy (OR: 3.06, 95% CI 1.62–5.78, *p* < 0.001) in initial H-B grades 5–6 were associated with more favorable outcomes at 6 months than with initial H-B grades 3–4. Our results showed that the proportion of patients who achieved favorable outcomes at 6 months and multiple clinical factors affecting favorable outcomes differed significantly among patients differing in initial severity of Bell’s palsy.

## 1. Introduction

Bell’s palsy is an idiopathic, acute peripheral facial neuropathy, which presents as unilateral weakness or paralysis of the face [1]. Its spontaneous recovery rate is high, and proper management results in a good prognosis without sequelae [2]. Many studies have assessed prognostic factors of outcomes, including accompanying symptoms, underlying diseases, such as hypertension and diabetes, age, and the degree of degeneration of the facial nerve as determined electrophysiologically. In addition, the initial degree of facial paralysis has been regarded as one of the most important prognostic factors for recovery [3]. In clinical practice, the degree of initial facial paralysis is determined by the House-Brackmann (H-B) grade, with lower H-B grade (less severe initial facial paralysis) being associated with a better prognosis. Nonetheless, few studies to date have assessed clinical factors associated with favorable outcomes in patients subgrouped by the degree of initial paralysis or initial severity of facial weakness. We hypothesized that clinical factors influencing favorable outcomes would differ by initial H-B grade.

Pathophysiologically, Bell’s palsy is characterized by vascular distension, inflammation, and edema with ischemia of the facial nerve, suggesting that this condition is caused by inflammation and viral infection [4]. Its treatments have therefore included corticosteroids and antiviral agents, separately or together [5]. Although international guidelines, including those of the American Academy of Otolaryngology, have recommended oral steroids as the primary treatment of Bell’s palsy [6], many studies have yielded conflicting results, suggesting that the choice of steroids and/or antiviral agents should be based on the severity of paralysis and that combined treatment with a steroid and an antiviral agent is more effective than either alone, especially in patients with severe Bell’s palsy [7,8,9,10]. However, the recent Cochrane review on combination therapy for Bell’s palsy was negative on the use of the combination antiviral and corticosteroids, even though it revealed that initially severe cases had higher rates of recovery after combination therapy compared with corticosteroids alone or placebo therapy [11]. Therefore, the present study evaluated whether the clinical prognostic factors differed according to the different initial H-B grades. This study also assessed the association of initial H-B grade with favorable outcomes in patients with Bell’s palsy.

## 2. Materials and Methods

### 2.1. Study Population and Design

For this retrospective cohort study, we obtained data from patients visiting the Department of Otorhinolaryngology, Head and Neck Surgery, who were diagnosed with acute facial palsy from 1 January 2005 to 31 January 2021. We included all patients who were aged 19 to 73 years in our study. For the present analyses of the Bell’s palsy patients, three otolaryngologists with more than 20 years of experience in facial paralysis reviewed the medical records of 1723 patients who were hospitalized for the management of acute facial palsy. Patient baseline demographics and characteristics, including age, sex, previous medical history before initiating treatment, and findings of the otorhinolaryngological examination, were evaluated. Exclusion criteria consisted of the following: (1) central nervous system disorders; (2) recurrent Bell’s palsy; (3) facial nerve palsy due to neoplasms; (4) Ramsay–Hunt Syndrome; (5) pregnancy, bilateral facial palsy, and past history of facial palsy; (6) aged under 19 years; (7) unclear timing of the first onset or follow-up less than 6 months; (8) H-B grade 2 at the first visit; (9) patients who failed to undergo invasive tests, such as electrophysiologic tests; and (10) patients who had supportive care alone. Finally, a total of 1315 patients were included in this study. 

All patients were prescribed an ophthalmic ointment with meticulous eye care. Patients desiring rehabilitation medicine were allowed physical therapy while in the hospital. Steroid treatment consisted of oral prednisolone for 10 days at a dose of 1 mg/kg per day (maximum, 80 mg/day) for the first 4 days, followed by 60 mg/day on days 5 and 6, 40 mg/day on days 7 and 8, and 20 mg/day on days 9 and 10. Because the patients in this study were recruited over a 15-year period, the choice of antiviral agent was dependent on the date of recruitment. Initially recruited patients were treated with oral acyclovir (1000–2400 mg/day) for 5 days, whereas those recruited later were treated with famciclovir (750 mg/day) for 7 days. 

### 2.2. Assessment of Outcome and Clinical Factors

Functional disability was evaluated using the H-B grading system [12]. The initial severity of Bell’s palsy was assessed by three otolaryngologists with more than 20 years of experience in facial palsy. Patients were divided into two groups based on the initial degree of facial palsy: H-B grades 3–4 (mild to moderate) and 5–6 (severe). Outcomes at 6 months were also evaluated based on H-B grade, with favorable outcomes defined as H-B grades 1–2 and unfavorable outcomes as H-B grades 3–6 (Figure 1).

Demographic and clinical characteristics were obtained from patient records, including age, gender, prior underlying conditions (e.g., diabetes mellitus, controlled hypertension), and the results of electrophysiological tests. Patients were evaluated by electroneurography (ENoG) 4 or 5 days after the onset of symptoms, and all patients underwent needle electromyography (EMG) 2 weeks after the onset of facial paralysis. Results were reported as the percentage of the maximal amplitude on the affected side divided by the maximal amplitude on the normal side. A poor ENoG result was defined as a >90% loss in amplitude, whereas a good result was defined as a ≤90% loss [13]. The presence or absence of a blink reflex and needle EMG results were analyzed simultaneously, with outcomes classified as poor or good by the physical medicine and rehabilitation physicians. The absence of pathologic spontaneous fibrillation activity was defined as a good outcome, whereas the presence of abnormal spontaneous activity or the absence of volitional activity was classified as a poor outcome. Treatment methods were classified as oral steroids alone and combined with antiviral therapy. 

The primary outcome was predefined as the percentage of patients achieving favorable outcomes 6 months after the onset of facial paralysis, as determined by H-B grade. Favorable outcomes were defined as H-B grades of 1 (normal) to 2 (mild dysfunction, defined as slight weakness on close inspection), whereas unfavorable outcomes were defined as H-B grades 3–6 [14,15]. Secondary outcomes included clinical factors associated with favorable outcomes in patients with different initial H-B grades. Associations between clinical factors and favorable outcomes in each group were evaluated by multiple logistic regression analyses and interactions of subgroup analyses. The protocol of this retrospective study was approved by the Institutional Review Board of Our University Hospital (IRB No. 2019-07-065), which waived the requirement for written informed consent due to the retrospective nature of the study.

### 2.3. Statistical Analyses

Categorical variables were reported as a number (percentage) and compared by a Chi-square test, whereas continuous variables were reported as mean ± standard deviation and compared with the Wilcoxon rank-sum test. Odds ratios (OR) and 95% confidence intervals (CI) were calculated by multivariable logistic regression. The interaction of the initial H-B grade with outcomes was considered significant for *p*-values less than 0.05, and interactions of subgroup analyses were evaluated using the subgroup-defining variable (variable × initial H-B grade) with the initial H-B grades 3–4 as a reference. All statistical analyses were performed using SAS 9.4 software (SAS Institute Inc., Cary, NC, USA) with *p* < 0.05 defined as statistically significant.

## 3. Results

This retrospective cohort consisted of consecutive 1723 patients aged 19 to 73 years who were diagnosed with acute facial palsy from 1 January 2005 to 31 January 2021. After excluding 408 patients who did not meet the inclusion criteria, the remaining 1315 patients were divided into two groups according to the degree of initial facial weakness (Figure 1), with 1019 (77.5%) patients having initial H-B grades 3–4 and 296 (22.5%) patients having initial H-B grades 5–6.

Table 1 presents the distribution of various demographics, clinical findings, electrophysiological variables, treatment methods, and treatment outcomes of the two groups of patients with Bell’s palsy. Age, sex, electrophysiologic variables, underlying disease (controlled hypertension, diabetes), and treatment methods did not differ significantly in the two groups. Of the 1315 included patients, 1047 (79.6%) showed favorable outcomes H-B grades 1–2) at 6 months (Figure 2). The percentage of patients who achieved a favorable functional outcome at 6 months was significantly higher in the group with initial H-B grades 3–4 than with initial H-B grades 5–6 (845/1019 (82.9%) vs. 202/296 (68.2%), *p* < 0.001; Table 1, Figure 2). 

Table 2 presents the results of multivariable logistic regression analyses of factors independently associated with prognosis in the groups of patients with initial H-B grades 3–4 and 5–6. Factors associated with favorable outcomes in patients with initial H-B grades 3–4 included age 19–40 years (OR: 2.33, 95% CI 1.27–4.26), age 41–65 years (OR:1.64, 95% CI 1.03–2.62), and good EMG results after 2 weeks (OR: 3.59, 95% CI 2.48–5.21). Factors associated with favorable outcomes in patients with initial H-B grades 5–6 included good EMG results after 2 weeks (OR:3.25, 95% CI 1.83–5.80), control of accompanying hypertension (OR: 2.49, 95% CI 1.32–4.67), and combination treatment with steroids and antiviral agents (OR: 1.82, 95% CI 1.04–3.18). Evaluation of subgroup interactions using the subgroup-defining variable (variable × initial H-B grade) and initial H-B grades 3–4 as a reference showed that treatment with a combination of steroid and an antiviral agent was associated with favorable outcomes in patients with initial H-B grades 5–6 (OR: 3.06, 95% CI 1.62–5.78, Figure 3).

## 4. Discussion

This study assessed whether demographic and clinical factors, including age, sex, electrophysiologic test results, controlled hypertension, diabetes, and treatment methods, were associated with initial H-B grade. The initial severity of facial paralysis has been shown to be associated with outcomes ranging from 6 months to 6 years in patients with Bell’s palsy [3,16] and to be the main factor predicting health-related quality of life in these patients [17]. Facial paralysis is usually evaluated using the H-B grading system, which grades facial function in six steps, ranging from normal (H-B grade 1) to total paralysis (H-B grade 6). Although strict criteria define a favorable outcome as H-B grade 1, the present study set H-B grades 1–2 as indicating favorable outcomes, thus excluding patients with initial H-B grade 2. Our results indicated that recovery rates after 6 months were higher in patients with initial H-B grades 3–4 than with initial H-B grades 5–6 (82.9% vs. 68.2%). These results were in accordance with those of previous studies, which found that rates of recovery were higher in patients with initially lower H-B grades. 

Generally, older age is negatively correlated with the rate of recovery from Bell’s palsy [3]. Our results showed that younger patients with initial H-B grades 3–4 had better outcomes than those aged >66 years. Although age was not significantly associated with outcomes in patients with initial H-B grades 5–6, the odds ratio for favorable outcomes was higher in younger patients. In this study, the odds ratio with favorable outcomes according to the severity of initial paralysis differed according to age. Especially, it is noteworthy that the younger age has a positive effect on recovery in the mild to moderate palsy group. 

Uncontrolled hypertension was also shown to correlate positively with facial palsy [18,19,20], whereas controlled hypertension has been reported to correlate with favorable outcomes. Appropriate control of blood pressure with antihypertensive agents has been associated with favorable outcomes after the development of severe Bell’s palsy. The pathophysiology responsible for the association between hypertension and facial palsy has not yet been determined. An autopsy study of two patients found blood clots in the tympanic part of the facial nerve canal, suggesting that these clots may have been caused by hemorrhaging in the facial nerve canal [18,21]. Bell’s palsy may also be caused by direct pressure due to the dilatation of blood vessels in the facial nerve or by edema due to increased intracranial pressure [22]. Therefore, more severe edema of the facial nerve would result in greater axonal loss of the facial nerve, resulting in severe facial paralysis. Uncontrolled hypertension may exacerbate nerve edema and direct pressure caused by the dilatation of blood vessels, resulting in severe palsy. Multivariable logistic regression analysis in our study showed that controlled hypertension was associated with a greater likelihood (odds ratio 2.49) of favorable outcomes in patients with severe facial palsy. Diabetes mellitus has been reported to contribute to poor prognosis in patients with Bell’s palsy. Another study found, however, that diabetes did not significantly affect the recovery rate [23]. Although our analysis indicated that the odds ratio for favorable recovery was higher in patients without than with diabetes, the absence of diabetes did not have a significant positive effect on recovery, regardless of initial facial severity. 

The results of electrophysiologic tests, including EMG and EnoG [24,25], have been shown to be useful in predicting the prognosis of patients with Bell’s palsy. Because this condition is accompanied by a conduction block and axonal degeneration, ENoG can evaluate the degenerated part of the axons [25]. Comparisons of the peak-to-peak amplitudes of the CMAP on the affected and unaffected sides can estimate the amount of nerve degeneration. The effects of treatment have been assessed separately in patients with good (ENoG ≥ 10%) and poor (ENoG < 10%) prognoses [26,27]. Abnormal spontaneous activity on needle EMGs, including positive sharp waves and fibrillation potentials, is regarded as indicating axonal degeneration, enabling more accurate predictions. Due to Wallerian degeneration, which generally occurs within 72 h after nerve injury, ENoG should not be performed prior to 3 days after nerve injury. ENoG after 7 days in patients with Bell’s palsy can reveal the extent of Wallerian degeneration [28]. In the present study, ENoG performed 4–5 days after the onset of Bell’s palsy did not yield clinically meaningful results due to Wallerian degeneration. In contrast, EMG performed 2 weeks after onset was predictive of prognosis, regardless of the initial severity of Bell’s palsy. Electrophysiologic results did not differ between patients with initially mild-to-moderate (H-B grades 3–4) and initially severe (H-B grades 5–6) palsy.

At present, patients with Bell’s palsy are treated with corticosteroids alone or corticosteroids plus antiviral agents. Oral corticosteroids are thought to reduce inflammation in patients with edema of the facial nerve, and antiviral agents are thought to eradicate herpes simplex virus infections [29]. The American Academy of Otolaryngology treatment guidelines has recommended that patients with Bell’s palsy be treated with oral corticosteroids alone within 72 h of symptom onset [6]. Although there is a consensus that the early use of corticosteroids is effective, the addition of antiviral agents was not beneficial for improving prognosis. Corticosteroids alone have been found to have a positive effect on recovery and prognosis [29,30]. However, previous studies of these treatment methods did not classify patients according to their initial degree of facial paralysis but only compared different treatment methods [2,31]. The inclusion of patients with different initial severities, including different mean H-B grades, has led to inconsistent results. Generally, outcomes are favorable in patients with mild to moderate palsy, even without treatment, due to the high spontaneous recovery rate [2]. The mixing of a large proportion of patients with mild to moderate palsy and a smaller proportion of patients with severe palsy may skew outcomes. The finding that a large proportion of patients with mild paralysis recovered spontaneously may have masked the positive effects of antiviral agents. The addition of antiviral agents to steroids resulted in a higher recovery rate than steroids alone [7], indicating the importance of determining initial severity when treating patients with idiopathic facial palsy. The present study found that the use of antiviral agents in addition to corticosteroids resulted in a higher likelihood (odds ratio 3.06) of favorable outcomes in patients with severe (H-B grades 5–6) than with mild to moderate (H-B grades 3–4) palsy. In contrast to previous studies, which reported that the addition of antiviral agents did not have beneficial effects in patients with Bell’s palsy (all grades), our results suggest that the addition of antiviral agents to corticosteroids may benefit patients with severe, but not with mild to moderate, Bell’s palsy at initial presentation. Double-blind, placebo-controlled, randomized trials are needed to compare the effectiveness of treatment according to the initial severity of Bell’s palsy. 

## 5. Strengths and Limitations

Most prior studies were meta-analyses that included large numbers of subjects. Therefore, inter-variability in the assessment process may be higher due to the differences in treatment agents and dosages and the duration of follow-ups. One of the strengths of this single-center study was its assessment of prognostic factors in patients differing to initial severity. To our knowledge, this is the first to classify different prognostic factors according to the initial severity of facial palsy and compare their interactions in patients with Bell’s palsy. In addition, few studies have been conducted on outcomes of Bell’s palsy with a large number of patients assessed for more than 15 years. The present study had several limitations. First, this study was non-randomized, which may have resulted in a high risk of bias due to confounders and treatment selection, which may have affected the effectiveness of treatment. Furthermore, although the initial severity of Bell’s palsy was assessed by three otologists, each with more than 20 years of experience in facial palsy, these evaluators were not blinded, which may have introduced selection biases. Second, ENoG was performed 4–5 days after symptom onset. Unfortunately, ENoG testing was not performed at the proper time, which may have affected patient outcomes. Third, facial function in these patients was assessed using the H-B grading system. However, synkinesis, a risk factor for poor prognosis and one of the major complications of Bell’s palsy, was not evaluated. These limitations may be overcome by high-quality randomized controlled trials. 

## 6. Conclusions

This large retrospective cohort of patients with Bell’s palsy found that the rate of favorable recovery at 6 months was higher in patients with initially mild to moderate (H-B grades 3–4) than initially severe (H-B grades 5–6) Bell’s palsy (82.9% vs. 68.2%). Multiple clinical factors affecting favorable outcomes differed depending on the initial severity of facial weakness: age younger than 66 years, and good EMG results (initial H-B grades 3–4), good EMG results, controlled hypertension, and combination antiviral treatment (initial H-B grades 5–6). In addition, there appeared to be an additional prognostic benefit to combination antiviral therapy only for those patients with initially severe than with initially mild to moderate Bell’s palsy.

## Figures and Tables

**Figure 1 jcm-10-03914-f001:**
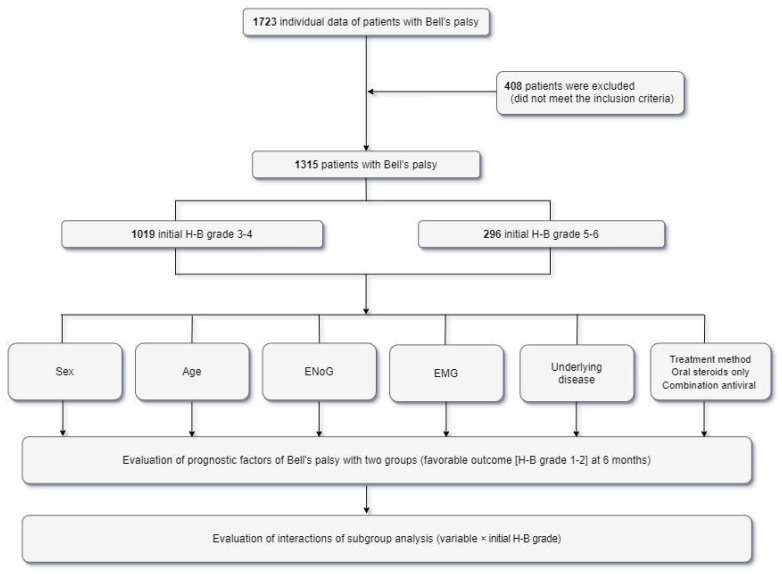
Flow diagram of the study population. H-B grade = House–Brackmann grade, ENoG = electroneuronography, EMG = electromyography.

**Figure 2 jcm-10-03914-f002:**
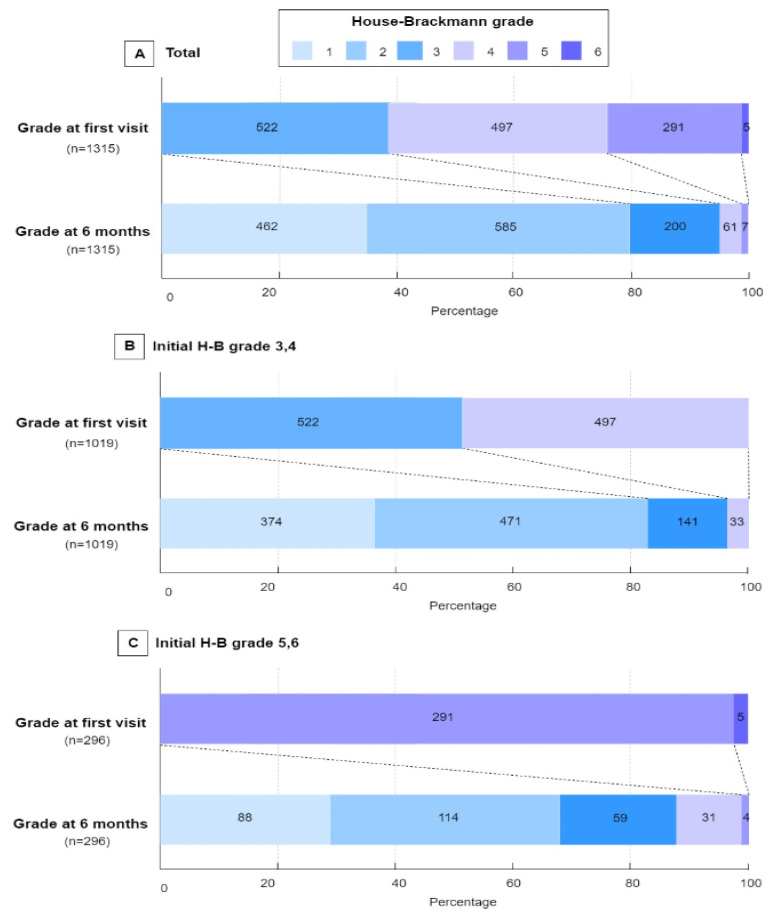
Distribution of the House–Brackmann grade at initially and at 6 months. The House–Brackmann (H-B) grade ranges from 1 (indicating normal) to 6 (total paralysis). Each cell corresponds with an H-B grade; the width of the cell represents the proportion of patients with equivalent grades and the number of patients is displayed within the cell.

**Figure 3 jcm-10-03914-f003:**
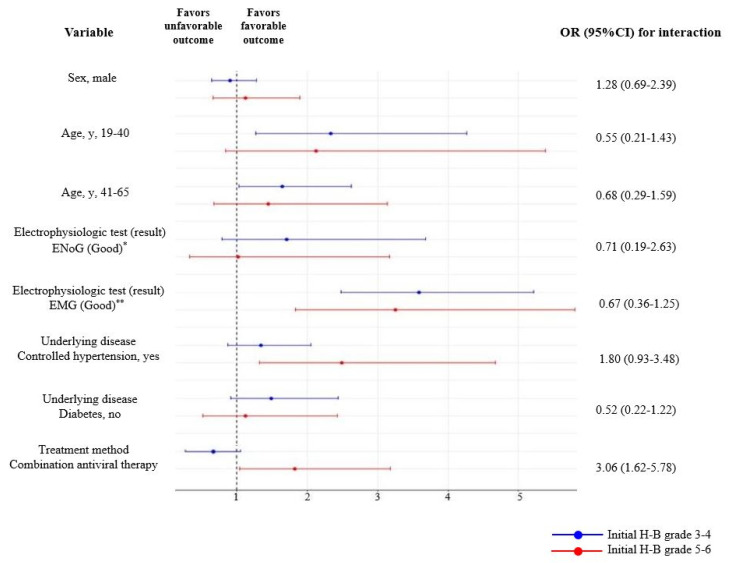
Results of subgroup analysis of secondary outcomes according to the initial H-B grade. The blue lines represent groups with initial House–Brackmann (H-B) grades 3–4, and the red lines represent groups with initial H-B grades 5–6. * Good results were defined as <90% degeneration on the affected side of the facial nerve, ** Good results were defined as the absence of pathologic spontaneous fibrillation activity. 95% CI = 95% confidence interval, H-B grade = House–Brackmann grade, ENoG = electroneuronography, EMG = electromyography.

**Table 1 jcm-10-03914-t001:** Patients’ demographic and clinical characteristics stratified by initial H-B grade.

Variable		*n*	H-B Grades 3–4 (*n* = 1019)	H-B Grades 5–6 (*n* = 296)	*p*-Value
Sex					
	Male	616	477 (46.8)	139 (47.0)	0.96
	Female	699	542 (53.2)	157 (53.0)	
Age group (years)					
	First tertile (19–40)	361	273 (26.8)	88 (29.7)	0.39
	Second tertile (41–65)	765	603 (59.2)	162 (54.7)	
	Third tertile (≥66)	189	143 (14.0)	46 (15.6)	
Electrophysiologic test (result)					
	ENoG (Good) *	1240	961 (94.3)	279 (94.3)	0.97
	ENoG (Poor)	75	58 (5.7)	20 (5.7)	
	EMG (Good) **	898	697 (68.4)	201 (67.9)	0.87
	EMG (Poor)	417	322 (31.6)	95 (32.1)	
Underlying disease					
	Controlled hypertension				
	Yes	823	643 (63.1)	180 (60.8)	0.47
	No	492	376 (36.9)	116 (39.2)	
	Diabetes				
	No	1132	881 (86.5)	251 (84.8)	0.47
	Yes	183	138 (13.5)	45 (15.2)	
Treatment method					
	Oral steroids only	623	474 (46.5)	149 (50.3)	0.25
	Combination antiviral therapy	692	545 (53.5)	147 (49.7)	
Outcome					
	Favorable ***	1047	845 (82.9)	202 (68.2)	<0.001
	Unfavorable	268	174 (17.1)	94 (31.8)	

H-B grade = House–Brackmann grade, ENoG = electroneuronography, EMG = electromyography. Data are presented as No. (percentage) of patients. * Good results were defined as <90% degeneration on the affected side of the facial nerve, ** Good results were defined as the absence of pathologic spontaneous fibrillation activity, *** Favorable outcomes were defined as an H-B grade of 2 or lower at 6 month after onset.

**Table 2 jcm-10-03914-t002:** Results of multivariable logistic regression analysis of favorable outcome stratified by initial H-B grade.

Variable		Initial H-B Grades 3–4	Initial H-B Grades 5–6
		Odds Ratio (95% CI)	Odds Ratio (95% CI)
Sex			
	Male	0.90 (0.64–1.28)	1.12 (0.66–1.89)
	Female	1 (Reference)	1 (Reference)
Age group (years)			
	First tertile (19–40)	2.33 (1.27–4.26)	2.12 (0.84–5.38)
	Second tertile (41–65)	1.64 (1.03–2.62)	1.45 (0.67–3.14)
	Third tertile (≥66)	1 (Reference)	1 (Reference)
Electrophysiologic test (result)			
	ENoG (Good) *	1.71 (0.79–3.68)	1.02 (0.33–3.17)
	ENoG (Poor)	1 (Reference)	1 (Reference)
	EMG (Good) **	3.59 (2.48–5.21)	3.25 (1.83–5.80)
	EMG (Poor)	1 (Reference)	1 (Reference)
Underlying disease			
	Controlled hypertension		
	Yes	1.34 (0.87–2.05)	2.49 (1.32–4.67)
	No	1 (Reference)	1 (Reference)
	DM		
	Absent	1.49 (0.91–2.44)	1.12 (0.52–2.43)
	Present	1 (Reference)	1 (Reference)
Treatment method			
	Oral steroids only	1 (Reference)	1 (Reference)
	Combination antiviral therapy	0.73 (0.45–1.09)	1.82 (1.04–3.18)

H-B grade = House-Brackmann grade, ENoG = electroneuronography, EMG = electromyography. * Good results were defined as <90% degeneration on the affected side of facial nerve, ** Good results were defined as the absence of pathologic spontaneous fibrillation activity.

## Data Availability

The data presented in this study are available on request from the corresponding author. The data are not publicly available as collected by using clinical records.

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
