# Peer review of "Association between Initial Severity of Facial Weakness and Outcomes of Bell’s Palsy"

_jcm, 2021, doi:10.3390/jcm10173914_

Round 1

Reviewer 1 Report

The large patient cohort presented in this manuscript makes it very interesting. To provide data over such a long time period with a comparable set of clinical tests and evaluations is outstanding. I was most impressed that all patients got the ENoG and the EMG at the same time point over years. Unfortunately, the timing for the ENoG was sub-optimal, what the authors discuss. 

The main criteria for the analysis was the initial visual grading using the House-Brackmann Facial Grading System. This is a common tool, but by far not optimal. Using more precise Grading Systems like the Sunnybrook Facial Grading System should be at least discussed. Also, the process of the grading should be explained in more details. Where the patients only graded face to face and only the grading documented? Or were photos taken and later reviewed? Were other grading systems used and do they bring more details information?

Could the authors differentiate in the 6-month outcome in House-Brackmann 1 and 2 instead of merging this two groups? It makes a big difference, if no sequel at all or at least a bit of facial palsy remains for the rest of the patients life. 

Have you done any otological test like measurement of stapedius reflexes? Volk et al have shown a strong positiv prognostic factor. (Volk GF, Klingner C,
Finkensieper M, et al. Prognostication of recovery time after acute peripheral
facial palsy: a prospective cohort study. BMJ Open 2013;3:e003007.
doi:10.1136/bmjopen-2013-003007 )

Reviewer 2 Report

The authors present a retrospective cohort study of a singular department in South Korea on the course of idiopathic peripheral facial nerve palsy. A total of 1723 patients' medical records were reviewed. After defined exclusion criteria, 1315 patients could be included in the study. In addition to supportive therapy (corneal care, rehabilitative measures), the patients received steroid treatment in descending doses and aciclovir or famciclovir as a virustatic agent. The severity of the facial paresis was classified according to Haus-Brackmann. The patients were divided into 2 groups, depending on the severity of the Haus-Brackmann scale. H-B grade III-IV (moderate to moderate) and H-B grade V-VI (severe). The outcome parameter evaluated was H-B grade at 6 months, with a good outcome defined as H-B grade I-II and an unfavourable outcome as H-B grade III-VI. The authors found a favourable outcome in 82.9% of patients with facial nerve palsy of initial grade III-IV. In the group of patients with facial nerve palsy of initial H-B grade V-VI, the outcome was favourable in 68.2%. This difference was statistically significant. In the multivariable logistic regression analysis, there was a significant advantage in the group of patients with H-B grade III-IV, for the age groups 19-40 years and 41-65 years, and for the group with good EMG findings after 2 weeks. For the group of patients with initial H-B grade V-VI, good EMG findings after 2 weeks, control of concomitant hypertension and combination therapy of steroids and viral drugs were positive factors. Evaluation of the group interaction showed that the combination of steroids and viral drugs was associated with a favourable outcome in the H-B grade V-VI patient group. In the discussion, the authors highlight that, in contrast to previous studies, the administration of a virustatic agent in patients with severe facial nerve palsy resulted in a significant positive effect. The strengths of the study are the high number of cases, the extensive and constant work-up, the long observation period and relatively uniform evaluation criteria.

Limitations

Page 6, lines 4 and 5, the line numbering breaks off here.

The sentence "Patients receiving....." is not clear and should be clarified.

The style of the bibliography is very strange. The Vancouver style is recommended
